# MSK1 Downstream Signaling Contributes to Inflammatory Pain in the Superficial Spinal Dorsal Horn

**DOI:** 10.3390/ijms262412177

**Published:** 2025-12-18

**Authors:** Jahanzaib Irfan, Rizki Muhammad Febrianto, Angelina Mira D’Ercole, Nicole Li, Vijaya Danke, Erica Chen, Deemah Aldossary, Michelle Y. Meng, Paolo La Montanara, Jose Vicente Torres-Perez, David Zimmermann, Rosalie Li, Krisztina Deak-Pocsai, Daniel Segelcke, Bruno Pradier, Esther Miriam Pogatzki-Zahn, Simone Di Giovanni, Michaela Kress, Istvan Nagy

**Affiliations:** 1Nociception Group, APMIC, Department of Surgery and Cancer, Imperial College London, Hammersmith Hospital, Du Cane Road, London W12 0NN, UK; j.irfan18@alumni.imperial.ac.uk (J.I.); febrianto.muhammadrizki@gmail.com (R.M.F.); angelina.mira01@gmail.com (A.M.D.); nicole.li@dpag.ox.ac.uk (N.L.); v.danke21@imperial.ac.uk (V.D.); d.aldossary21@imperial.ac.uk (D.A.); michelle.meng@yahoo.co.uk (M.Y.M.); montanara03@gmail.com (P.L.M.); rosalieli@hotmail.com (R.L.); 2Division of Neuroscience, Department of Brain Sciences, Imperial College London, Hammersmith Hospital, Du Cane Road, London W12 0NN, UK; s.di-giovanni@imperial.ac.uk; 3Department of Cellular Biology, Functional Biology and Physical Anthropology, Faculty of Biological Sciences, University of Valencia, C/Dr. Moliner 50, 46100 Burjassot, Spain; jose.vicente.torres@uv.es; 4Institute of Physiology, Medical University of Innsbruck, 6020 Innsbruck, Austria; david.zimmermann@i-med.ac.at (D.Z.); michaela.kress@i-med.ac.at (M.K.); 5Department of Physiology, University of Debrecen, 98 Nagyerdei krt, 4012 Debrecen, Hungary; deak-pocsai.krisztina@med.unideb.hu; 6Department of Anaesthesiology, Intensive Care and Pain Medicine, University Hospital Muenster, 48149 Muenster, Germany

**Keywords:** spinal cord, neuron, glia, nociceptive processing

## Abstract

The nuclear kinases mitogen- and stress-activated kinase 1 and 2 (MSK1 and MSK2), through regulating transcriptional processes, are pivotal for various adaptive responses, including inflammation, learning and addiction. Transcriptional alterations in neurons and glia cells within the pain signal-processing (nociceptive) pathway, including the superficial spinal dorsal horn (SSDH), are critical for the development and persistence of inflammatory pain that results from tissue injuries and subsequent inflammatory reactions. While previous reports have indicated that MSK1 contributes to transcriptional changes in inflamed tissues, the impact of MSK1 on nociceptive processing in the SSDH are poorly understood at present. Here, we report MSK1 immunoreactivity (IR) in a group of excitatory and inhibitory neurons as well as in microglia and oligodendrocytes in the SSDH. Injecting Complete Freund’s Adjuvant into the mouse hind paw produced robust non-evoked pain-related behavior, which was significantly attenuated by global depletion of MSK1. In wild-type mice, the inflammatory pain was accompanied by transient MSK1-dependent phosphorylation of the MSK1 downstream effector histone 3 at serine 10 at one hour but not two days after the injection; still, the number of nuclei exhibiting activated MSK1 expression remained highly restricted even at 1 h post-injection. Our data indicate that MSK1 contributes to inflammatory pain via epigenetic and transcriptional alterations; however, once initiated, MSK1’s downstream effects do not require further drive from the persistent activity of the MSK1 signaling pathway in the SSDH.

## 1. Introduction

Pain characterized by hypersensitivity to mechanical and heat stimuli (i.e., mechanical allodynia and heat hyperalgesia, respectively) is one of the cardinal signs of inflammation [1]. Inflammatory pain lasts until healing nears completion and often beyond [2]. The development and persistence of inflammatory pain depend on adaptive changes, which comprise mechanistically distinct components, such as self-amplification, post-translational modifications and transcriptional alterations of a series of membrane and cytoplasmic molecules, leading to a sensitized state that is characterized by increased responsiveness and activity of neurons all along the pain signal-processing (nociceptive) pathway [3,4,5]. While the transcriptional component of sensitization is particularly important for the persistence of inflammatory pain [6], our current understanding of the involved regulatory mechanisms and the genes affected by the regulators is limited.

The nuclear mitogen- and stress-activated kinases (MSK1 and MSK2) are involved in the initiation and maintenance of various adaptive changes, including inflammatory reaction, learning and addiction, through the regulation of transcriptional processes, both within and outside the nervous system [7,8,9,10]. Evidence of the roles of MSK1 and MSK2 in the development and persistence of pain is emerging; global depletion of both MSK isoforms specifically inhibits the development of inflammatory heat hyperalgesia, while MSK1 contributes to the sustained component of pain evoked by activation of chemosensitive primary sensory neurons by formalin and developing after peripheral sensory neuropathy [11,12,13,14].

In the central nervous system (CNS), the superficial spinal dorsal horn (SSDH) is the first site for integrating nociceptive information, and it determines the quality and quantity of sensory information forwarded to higher centers of the nociceptive system [3,15,16]. Information processing in the SSDH is executed in neuronal circuitries involving presynaptic terminals of primary sensory neurons targeting excitatory and inhibitory spinal neurons and influenced by crosstalk between neurons, glia cells and inhibitory and excitatory pathways descending from brain stem nuclei [3,15,16]. MSK1 plays a relevant role in nociceptive signal processing in the SSDH; for example, formalin injection into the mouse hind paw, as well as peripheral neuropathy, activates MSK1 in a group of SSDH cells, and an intrathecally applied MSK1 inhibitor has attenuated pain behavior in both pain models [11,13]. Further, tissue injury, activation of primary sensory neurons by noxious stimuli and peripheral neuropathy up-regulate MSK1-dependent expression of the epigenetic tag phosphorylated (p) serine 10 (S10) of histone 3 (H3; p-S10H3) in the SSDH [11,12,13,17,18]. These findings together show that MSK1 could significantly contribute to nociceptive processing in the SSDH. However, MSK1 expression and signaling in the SSDH, in models of tissue injury or inflammation, are not sufficiently understood.

Therefore, we characterized MSK1 protein expression and assessed the role of MSK1 in nociceptive processing in the SSDH in peripheral inflammation. We report that MSK1 was expressed in populations of excitatory and inhibitory neurons, microglia and oligodendrocytes, and that peripheral inflammation-activated signaling cascades downstream of MSK1.

## 2. Results

### 2.1. MSK1 Expression in SSDH Neuron, Microglia and Oligodendrocyte Sub-Populations

MSK1-expressing nuclei were detected both in the white and gray matter of the spinal cords of WT but not of MSK1^−/−^ mice (Figure 1A; Appendix A). Combined immunostaining showed MSK1-expressing nuclei predominantly in the cells, which were immunopositive for the neuronal nucleus marker hexaribonucleotide-binding protein-3 (NeuN), the microglia marker ionized calcium-binding adapter molecule 1 (IBA1) or the oligodendrocyte marker 2′,3′-Cyclic-nucleotide 3′-phosphodiesterase (CNPase), suggesting that MSK1 was present in the sub-populations of neurons, microglia and oligodendrocytes (Figure 1A–H). However, MSK1 expression was rarely detected in cells expressing immunoreactivity for the astrocyte marker glial fibrillary acidic protein (GFAP) (Figure 1I–K).

Overall, one-fourth of the nuclei exhibited MSK1 immunopositivity in laminae I-II (Figure 2; Appendix A). More than two-thirds of the MSK1-expressing nuclei belonged to neurons located in laminae I and II (Figure 2; Appendix A). Among the neurons, inhibitory neurons immunoreactive to paired box gene 2 (PAX2) and excitatory neurons immunoreactive to LIM homeobox transcription factor 1-beta (LMX1-β) exhibited MSK1 expression in the nucleus (Figure 3; Appendix A). Although MSK1 expression was higher among the PAX2- than the LMX1-β-immunoreactive (IR) neurons, about the same proportion of all inhibitory and excitatory neurons (either LMX1-β-IR or no IR for either LMX1-β or PAX2) exhibited MSK1-IR (Figure 2 and Figure 3; Appendix A). Regarding other cell types, almost half of the microglia and oligodendrocyte nuclei expressed MSK1 (Figure 1 and Figure 2; Appendix A), whereas only very few astrocytes in laminae I–II showed MSK1-IR in the nucleus (Figure 1 and Figure 2; Appendix A). A series of quadruple staining confirmed the cellular pattern of MSK1 expression described above (Appendix A).

### 2.2. MSK1-Dependent Signaling in SSDH by Peripheral Inflammation

S10H3 is one of the downstream effectors for MSK1, which is up-regulated in the SSDH following injury or inflammation of peripheral tissues [12,18]. To investigate whether inflammation of the hind paw led to MSK1 signaling in the SSDH, we injected Complete Freund’s Adjuvant (CFA) into the hind paw of wild-type (WT) and MSK1 global knockout (MSK1^−/−^) mice and used p-S10H3 as a read-out. CFA injection, as expected, induced inflammatory pain indicated by the significant reduction in the ipsilateral–contralateral footprint ratio in both genotypes (Figure 4). However, the MSK1^−/−^ mice exhibited reduced pain-related behavior, indicating that MSK1 significantly contributes to inflammatory pain.

CFA injection also induced the expression of a relatively high number of p-S10H3-IR nuclei in the ipsilateral SSDH one hour after CFA injection in WT mice (21.5 ± 3.9 neurons/section, n = 3). However, p-S10H3-expressing nuclei were barely detectable two days after CFA injection (0.6 ± 0.4 neurons/section, n = 3). Further, p-S10H3-IR was not observed in naïve mice or CFA-injected MSK1^−/−^ mice one hour after the injection (Figure 5A–D; Appendix A).

In spite of the critical role of MSK1 in H3 phosphorylation at S10 shown by the lack of p-S10H3 expression in CFA-injected MSK1^−/−^ mice, and in contrast to the significant number of p-S10H3-expressing nuclei in the SSDH, very few nuclei exhibited p-MSK1-IR in the ipsilateral SSDH one hour after CFA injection (1.4 ± 0.9 neurons/section, n = 3; Figure 6A,C). No p-MSK1-expressing nuclei were detected in the SSDH two days after CFA injection in WT mice (Figure 6B,C), and no p-MSK1-IR-expressing cells were encountered in naïve WT mice (Appendix A). These data strongly suggest a critical role of the transient activation of MSK1 signaling in SSDH neurons and possibly also non-neuronal cells following peripheral inflammation.

## 3. Discussion

In agreement with previous reports and data emerging from single-cell/single-nucleus transcriptomics [11,12,13,19], we found that MSK1 was expressed in the spinal cord. Specifically in laminae I-II, almost 1/3 of the neurons and almost half of the microglia and oligodendrocytes expressed MSK1 in the nucleus. This expression pattern agrees with transcriptomic findings that indicate strong expression of *Rps6ka5*, the gene encoding MSK1, in mature oligodendrocytes and weaker expression in microglia as well as various types of neurons [19]. While transcriptomic data have also revealed *Rps6ka5* expression in astrocytes [19], our immunolabeling identified only a few astroglia with MSK1 expression in the nuclei, suggesting that *Rps6ka5* translation efficacy might be low in those glia cells, at least in the naïve condition.

Among neurons, we detected MSK1 expression in the nuclei of about 1/3 of the PAX2- and 1/8 of LMX1-β-expressing cells. While PAX2 is expressed in all inhibitory neurons, LMX1-β is expressed in only a proportion of excitatory cells [20,21,22,23,24,25]. Hence, MSK1-expressing neurons, which lack PAX2 and LMX1-β expression, are likely excitatory cells. The MSK1 expression pattern is in line with recent transcriptomic data indicating *Rps6ka5* mRNA expression in some inhibitory neuron clusters, particularly in type-1, -2, -9 and -11 clusters, as well as in excitatory neurons [19].

MSK1 contributes to pain that follows tissue injury [11,14]. Here, evidenced by reduced non-evoked pain-related related behavior and MSK1-dependent up-regulation of p-S10H3 expression, which is an epigenetic marker of nociceptive processing in the SSDH [11,12], we demonstrated that MSK1 also contributes to inflammatory pain. MSK1-dependent up-regulation of p-S10H3 in the SSDH occurs within minutes following stimulation of nociceptive primary sensory neurons with irritants, such as capsaicin or formalin, and tissue injury, such as burn injury [11,12,18]. Inhibition of S10H3 phosphorylation in dynorphin-containing SSDH neurons, which constitute the major sub-population of cells expressing MSK1-dependent p-S10H3 after burn injury, increases the thermal pain threshold without alteration in mechanical pain threshold [18,26]. Formalin injection into the rat hind paw also increases the expression of p-MSK1, and intrathecal injection of the MSK1/2 inhibitor SB747651-A significantly inhibits H3 phosphorylation at S10 and attenuates second-phase pain-related behavior associated with formalin injection [11]. Together, these findings suggest that MSK1 activation in the SSDH is critically involved in the development and persistence of inflammatory pain and heat hyperalgesia. However, in the present study, we found only a few p-MSK1-expressing nuclei in the entire spinal dorsal horn one hour after CFA injection into the hind paw and no p-MSK1-expressing nuclei two days after the CFA injection. Yet a significant number of p-S10H3-expressing nuclei were detected at one hour but not at two days after CFA injection. While the lack of p-S10H3 two days after CFA injection in the spinal cord agrees with the rapid downregulation of the epigenetic tag, the very low p-MSK1 expression is highly surprising, since MSK1 acts as a writer for p-S10H3 (present study; [4,10,11,12]).

The highly restricted expression of p-MSK1 IR nuclei could be due to at least two reasons. First, MSK1 phosphorylation–dephosphorylation could occur within a short time after the inflammatory signal reaches the spinal cord, and pMSK1 becomes undetectable by 1 h after CFA injection. This possibility is supported by the finding that the formalin injection-induced p-MSK1 expression is present in the spinal dorsal horn 30 min after injection [11]. Second, the amount of phosphorylated MSK1 molecules could be below the detection limit of the immunoreaction in the majority of nuclei after CFA injection at any time point. This might be related to the different magnitudes of nociceptive input from primary nociceptive afferents to neurons and glia in the SSDH at any time point during the initial hours after the respective injections; formalin immediately activates nociceptors, supplying the injection site and expressing the transient receptor potential ion channel subfamily A member 1 (TRPA1), resulting in a significant and sudden increase in spinal nociceptive input [27], whereas CFA gradually induces inflammation, resulting in gradually increasing nociceptive input into the spinal cord [28]. Nevertheless, the rapid downregulation of p-S10H3 expression after both burn injury and CFA injection suggests that regardless of the persisting peripheral inflammation, activation of MSK1’s upstream effectors could be also highly transient in the SSDH.

MSK1 regulates a series of genes including the immediate early gene *fos* [29,30], the neuropeptide Y-encoding *Npy* [31], the interleukin 1b and 6 genes *Il1b* [32] and *Il6* [33] and the gene-encoding tumor necrosis alpha *Tnfa* [32,34], which can significantly contribute to the modulation of spinal nociceptive processing [14]. Based on the temporal dynamics of p-MSK1 and p-S10H3 expression and the apparent role of MSK1 in the persistence of inflammatory pain in the SSDH [11], we propose that transcriptional regulation of MSK1-dependent genes, including via epigenetic mechanisms involving p-S10H3, develops immediately after injury but does not require further drive for persistence from MSK1 activity. Therefore, MSK1 may act as a transient initiator rather than a persisting decisive switch in the SSDH for the development of heat hyperalgesia as a consequence of injury or inflammation of peripheral tissues.

## 4. Materials and Methods

### 4.1. Animals

Mice with a global depletion of MSK1 (MSK1^−/−^; C57BL6) and their wild-type (WT) littermates (20–28 g; male/female ratio ~50%) were used in this study. The animals were kept in Imperial College London’s Central Biological Services unit in humidity- and temperature-controlled rooms at a 12 h light–dark cycle and with access to food and water *ad libitum*. WT and MSK1^−/−^ mice were separately housed in enriched environments in small mouse IVCs (1–3 mice/cages). Each cage contained either naïve or CFA-injected mice. The animals were accustomed to the experimental person and handled by tunnel.

All animal experiments were carried out in accordance with the UK Animals (Scientific Procedures) Act of 1986; the relevant European Communities Council Directive (86/609/EEC); the guidelines of the Committee for Research and Ethical Issues of ISAP and the National Institutes of Health *Guide for the Care and Use of Laboratory Animals* (Revised Guidelines). The Animal Welfare and Ethical Review Body, Imperial College London, UK, approved all procedures on animals, and the animal work was conducted under a Home Office Project License (P5D499DB0, 9 January 2019). We also adhered to the Good Lab Practice and ARRIVE guidelines in this study.

### 4.2. Inflammatory Model and Tissue Harvesting

Naïve mice and mice 1 h or 2 days after receiving 25 µL CFA injected into one of the hind paws under isoflurane (3%) anesthesia were terminally anaesthetized by intraperitoneal pentobarbital and transcardially perfused with phosphate-buffered saline (PBS) followed by 4% paraformaldehyde (PFA) solution. The inflammatory reaction was ascertained by inspecting the paw for swelling and redness before terminal anesthesia. The spinal cord was dissected, immersed in 4% PFA solution for up to one hour and subsequently stored at 4 °C in 30% sucrose solution.

### 4.3. Behavioral Testing

In a group of WT and MSK1^−/−^ mice (using a 50% male–female ratio), we also assessed the effect of CFA injection on non-evoked pain-related behavior [35], which included putting mice into Perspex boxes positioned on a Perspex platform illuminated with green LED light. A camera was placed underneath the Perspex platform, and 5 min footage intervals of animals’ activities were shot. Individual images from the video depicting mice using the ipsilateral or contralateral paw for weight bearing were carefully selected, and areas of the ipsilateral and contralateral hind paw prints were measured and compared off-line using Fiji/ImageJ (Imagej.net, 2.14.0/1.54f). Paw prints for each animal were measured on at least 5 images, and the measurements were averaged. The ipsilateral paw print areas were then normalized to the contralateral, and the ratios were used for statistical analysis.

### 4.4. Immunostaining

Following cryoprotection, the lumbar (L) 3–5 spinal segments were isolated and embedded in OCT. Cryosections (20 µm) were prepared, positioned serially on Superfrost plus microscope slides (Thermo Fisher Scientific, Loughborough, UK) so that adjacent sections were separated by 200–300 μm and processed for immunostaining. Sections were blocked with 10% normal goat serum for one hour followed by incubation with primary antibodies diluted in PBS containing 0.3% Triton X (PBST) and 1% normal goat serum overnight. The slides were then incubated with Alexa Fluor-conjugated secondary antibodies diluted in PBST for one hour and covered in ProLong^TM^ Gold antifade reagent with DAPI (Thermo Fisher Scientific, Loughborough, UK). All incubations were performed in a humidified chamber at room temperature, and the slides were washed between incubations three times for 5 min with PBST.

When two primary antibodies that had been raised in the same species were employed for co-staining, the Biotin-XX Tyramide SuperBoost^TM^ Kit (Thermo Fisher Scientific, Loughborough, UK) was used according to the manufacturer’s instructions to amplify the signal generated by the first primary antibody. Briefly, following incubation with the first primary antibody, sections were incubated with a horseradish peroxidase-linked goat anti-rabbit secondary antibody for one hour. Tyramide working solution was applied for 5 min, then sections were incubated with Alexa-fluor-conjugated Streptavidin for one hour. Sections were incubated with the second set of primary and secondary antibodies, and subsequent steps were repeated as described above. Primary and secondary antibodies and normal sera are listed in Appendix A.

### 4.5. Imaging and Quantification

Immunostained sections were visualized using a HWF1–Zeiss Axio Observer inverted microscope with a fully motorized stage, controlled by Zen acquisition software. Images were captured with a Hamamatsu Flash 4.0 fast camera (Welwyn Garden City, UK). Fiji/ImageJ (Imagej.net, 2.14.0/1.54f) software was used to analyze and quantify the immunostaining.

For each animal, three to five 25 µm-thick spinal cord transverse sections at the L3–L5 level were examined. Borders between the white and gray matter and lamina II and lamina III were drawn on the image, and regions of interest (ROIs) were defined. Cell counts were obtained either with Fiji/ImageJ or manual counting to isolate individual cells. The cell numbers found in the sections of each animal were averaged and used for statistical analysis and visualization.

### 4.6. Statistical Analysis

Statistical analysis included checking data for normal distribution using skewness and a kurtosis z-values test. Normality was assumed if a z-value was within ±1.96 and a *p*-value > 0.05 in Shapiro–Wilk data analysis. Differences in immunopositive cell numbers were tested using ANOVA followed by the Bonferroni post-hoc test. Behavioral data were tested by two-way multiple-comparison ANOVA followed by the Bonferroni post-hoc test. Data are presented as means ± SEM (standard error of the mean). Differences were considered statistically significant at *p* < 0.05.

## Figures and Tables

**Figure 1 ijms-26-12177-f001:**
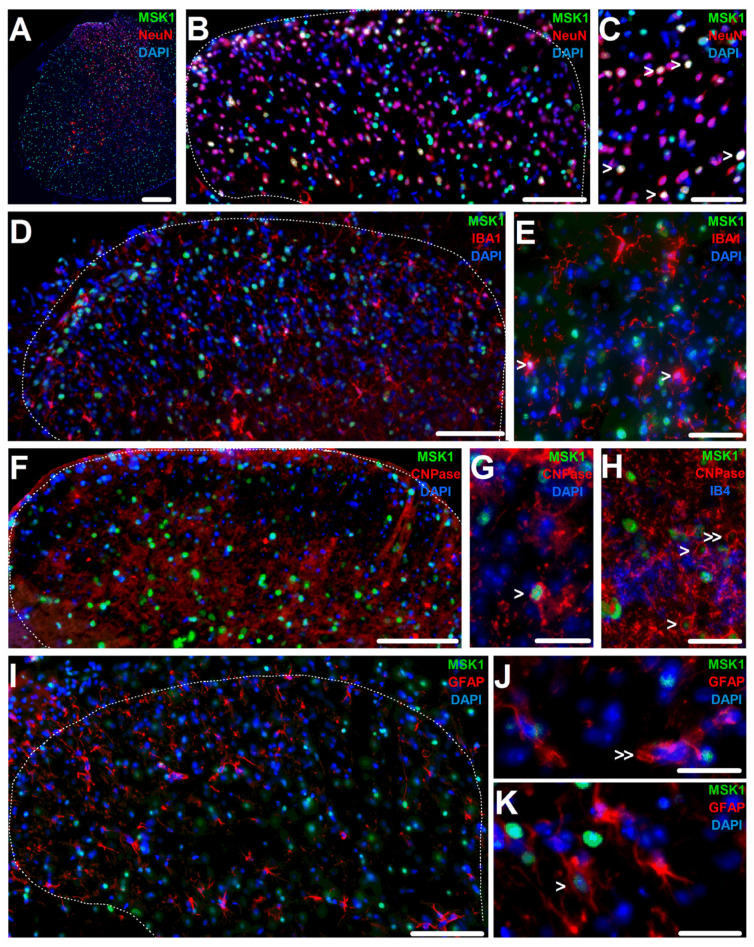
MSK1’s nuclear expression in sub-populations of neurons and glia cells in the mouse SSDH. (**A**) Microphotograph showing MSK1 expression in nuclei of neurons and non-neuronal cells in gray and white matter of the spinal cord. (**B**) Microphotograph of the spinal dorsal horn showing MSK1-expressing nuclei of neurons and non-neuronal cells. (**C**) High-magnification image of the dorsal horn depicting neuronal nuclei, a few indicated by arrowheads. (**D**) Microscopic image showing microglia exhibiting MSK1 expression in the cells’ nuclei in the SSDH. (**E**) High-magnification image of the SSDH showing microglia expressing MSK1 in the nucleus, some indicated by arrowheads. (**F**) Microphotograph of the SSDH showing oligodendrocytes with MSK1-expressing nuclei. (**G**,**H**) High-magnification images of the spinal dorsal horn depicting oligodendrocytes with (arrowhead) or without (double arrowhead) MSK1 expression in the cells’ nuclei. (**I**) Microscopic image of the SSDH showing astrocyte and MSK1 expression. (**J**,**K**) High-magnification microscopic images of the SSDH depicting astrocytes without ((**J**); double arrowhead) or with MSK1 expression ((**K**); arrowhead). The dotted line in (**B**,**D**,**F**,**I**) indicates border between white and gray matter. Scale bars = 250 μm in (**A**), 100 μm in (**B**,**D**,**F**,**I**), 50 μm in (**C**,**E**) and 25 μm in (**G**,**H**,**J**,**K**).

**Figure 2 ijms-26-12177-f002:**
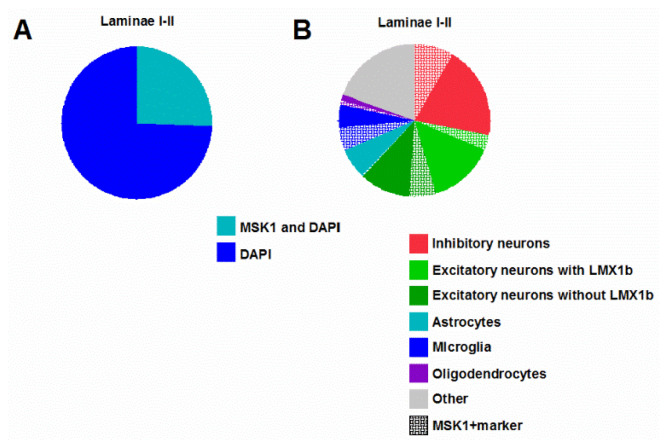
MSK1 expression in various cell types in the mouse SSDH. (**A**) About a quarter of the nuclei express MSK1 in the SSDH. (**B**) Proportion of various cell types with (hashed areas) or without (solid areas) MSK1-expressing nuclei in the SSDH.

**Figure 3 ijms-26-12177-f003:**
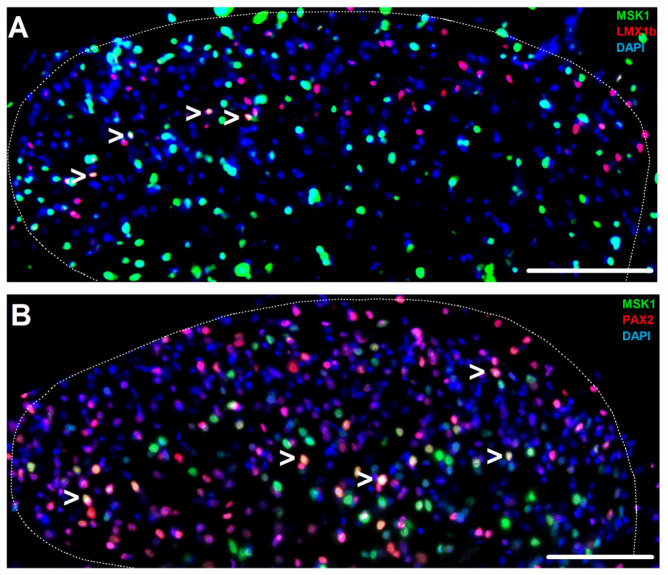
MSK1 expression in sub-populations of inhibitory and excitatory neurons in the mouse SSDH. (**A**) Microphotograph of the spinal dorsal horn depicting MSK1-expressing nuclei (arrowheads) in excitatory neurons. (**B**) Microscopic image of the spinal dorsal horn showing inhibitory neurons having MSK1-expressing nuclei (arrowheads). Dotted lines indicate the border between the white and gray matter in both images. Scale bar = 100 μm.

**Figure 4 ijms-26-12177-f004:**
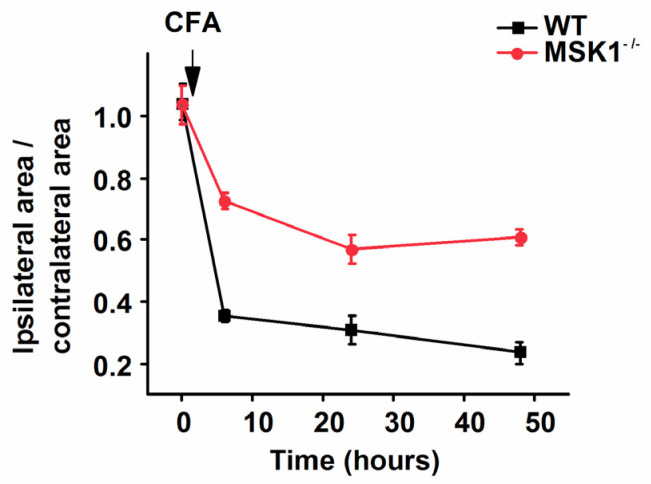
CFA injection-induced inflammatory pain is indicated by non-evoked pain-related behavior. This line chart depicts time-dependent significant change in the ipsilateral–contralateral footprint ratio both for WT and MSK1^−/−^ mice and significant attenuation of inflammatory pain by global MSK1 depletion. WT_baseline_ v MSK1^−/−^_baseline_ *p* = 1. WT_baseline_ v WT_CFA_ *p* < 0.0001 for each time point. MSK1^−/−^_baseline_ v MSK1^−/−^_CFA_ *p* < 0.0001. WT_CFA_ v MSK1^−/−^_CFA_ *p* = 0.0013—*p* < 0.0001. n = 4 for each genotype.

**Figure 5 ijms-26-12177-f005:**
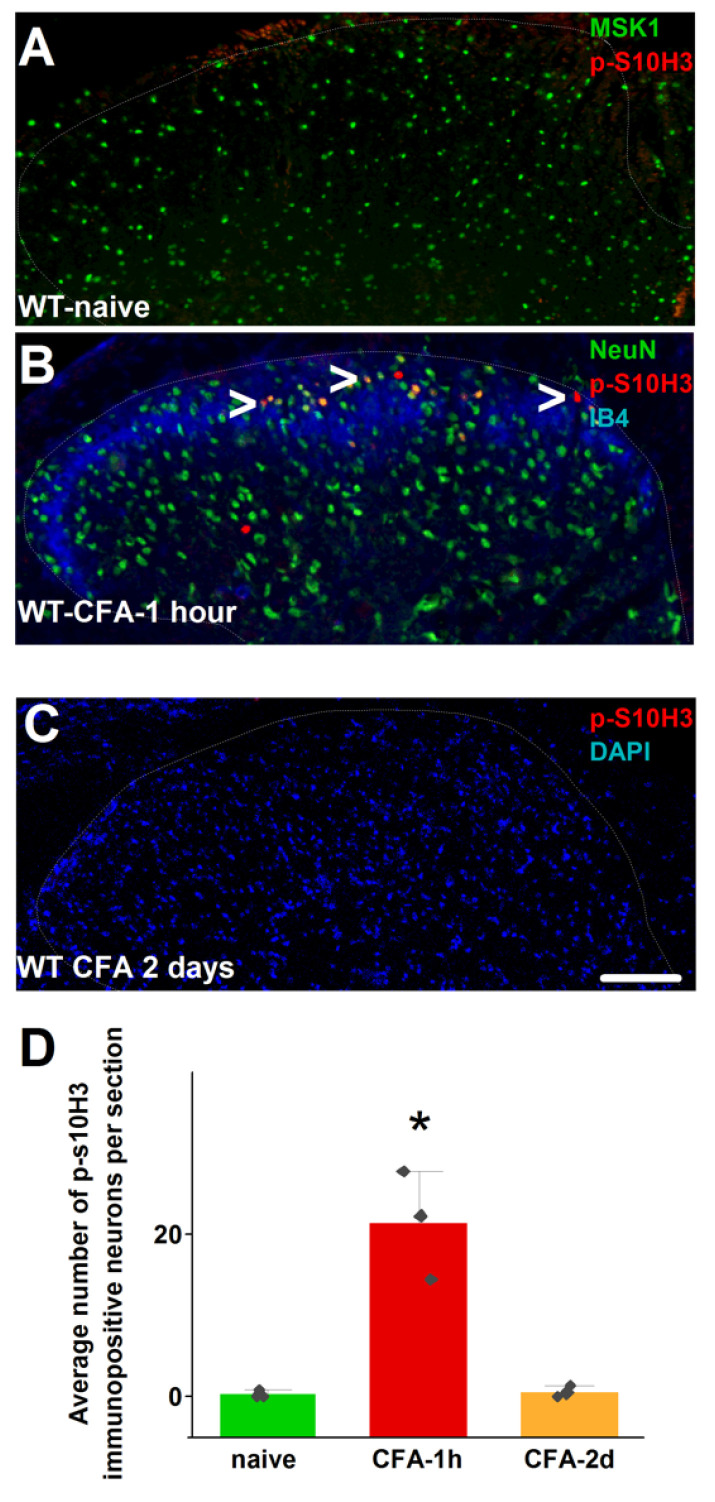
Phosphorylated serine 10 in histone expression in a WT mouse L4–5 SSDH. (**A**) Microscopic image of a naïve mouse’s SSDH showing no p-S10H3 expression. (**B**) Microphotograph of a mouse’s spinal dorsal horn 1 h after injecting Complete Freund’s Adjuvant (CFA) into one of the hind paws, showing p-S10H3-expressing neuronal nuclei, some indicated by arrowheads. (**C**) Microphotograph of a mouse’s spinal dorsal horn 2 days after CFA injection into the hind paw, showing no p-S10H3-expression. (**D**) Changes in the number of p-S10H3-expressing nuclei in the WT mouse L4–5 SSDH one hour and 2 days after CFA injection into the hind paw. The asterisk indicates a significant difference: *p* = 0.0017. Scale bar = 100 μm (all images).

**Figure 6 ijms-26-12177-f006:**
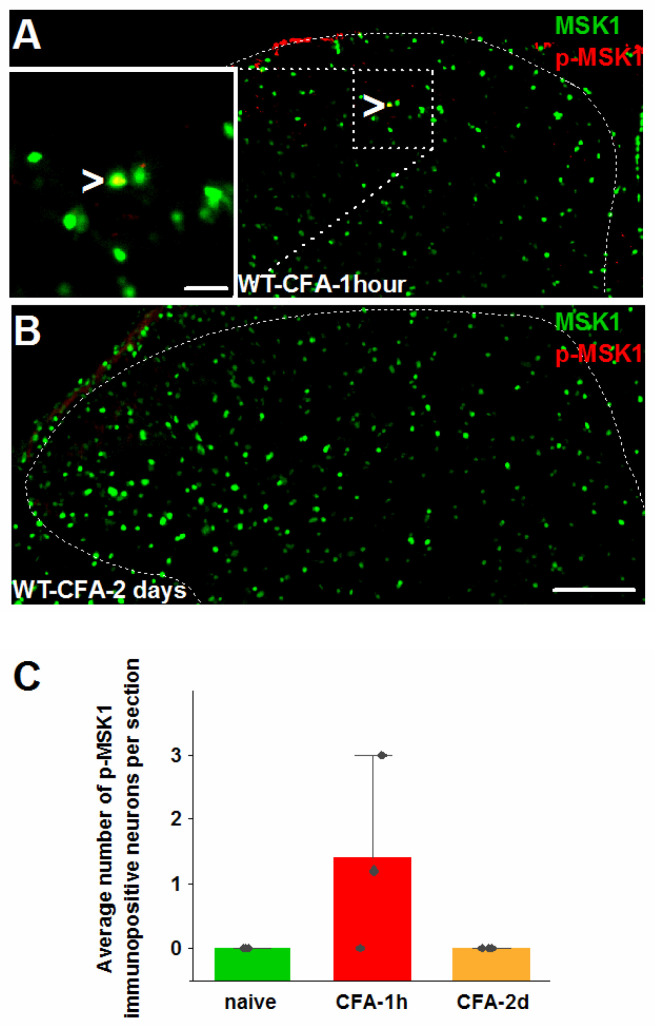
Phosphorylated MSK1 expression in WT mice L4–5 SSDH. (**A**) Microscopic image of the spinal dorsal horn showing an activated MSK1 (p-MSK1)-expressing nucleus one hour after CFA injection into the hind paw (arrowhead). (**B**) Microphotograph of the spinal dorsal horn collected 2 days after CFA injection into the hind paw, showing no p-MSK1 expression. (**C**) Changes in the number of p-MSK1-expressing nuclei in the L4–5 SSDH one hour and 2 days after CFA injection into the hind paw. Dotted lines indicate border between white and gray matter. Scale bar = 100 μm in low-magnification images and 20 μm in insert.

## Data Availability

All raw and processed data are available from I.N. following request.

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
