# Peer review of "MSK1 Downstream Signaling Contributes to Inflammatory Pain in the Superficial Spinal Dorsal Horn"

_ijms, 2025, doi:10.3390/ijms262412177_

Round 1
Reviewer 1 Report
Comments and Suggestions for Authors
The manuscript presents valuable data on the baseline expression of MSK1 in the superficial dorsal horn of the spinal cord. However, the second part of the paper presents results that are somewhat inconsistent.
The authors discuss MSK1 activation in response to CFA injection, but they do not demonstrate this activation in their present experiments or cite any previous studies that have confirmed it. Based on findings obtained in studies using other inflammatory models, the activation of S10H3 and the lack of MSK1 activation in MSK1 KO mice, the authors suggest that MSK1 may be activated over a short period (less than 1 hour) after CFA injection. It would be welcome if an early MSK1 upregulation could be revealed directly. In addition, the heading „MSK1 activation in SSDH by peripheral inflammation” should be reformulated.
In the title, inflammatory pain is mentioned, but no pain responses were measured in the present study. The title should be reformulated.
In the case of Figs. 4 and 5, no statistics are mentioned, and no significance is indicated.
I found no table in the manuscript, although the authors cite Tables 1-3.
Author Response
We are grateful for both referees for the constructive comments and criticisms. As you will note, during the revision we have accepted reviewers’ suggestions, which we believe significantly improved the quality of the manuscript. Please find our responses and brief description of our actions during the revision below.
Reviewer 1
The manuscript presents valuable data on the baseline expression of MSK1 in the superficial dorsal horn of the spinal cord. However, the second part of the paper presents results that are somewhat inconsistent.
The authors discuss MSK1 activation in response to CFA injection, but they do not demonstrate this activation in their present experiments or cite any previous studies that have confirmed it.
Agree. We do not demonstrate robust MSK1 activation, and no previous data are available about CFA injection-induced MSK1 activation in the spinal dorsal horn.
Based on findings obtained in studies using other inflammatory models, the activation of S10H3 and the lack of MSK1 activation in MSK1 KO mice, the authors suggest that MSK1 may be activated over a short period (less than 1 hour) after CFA injection. It would be welcome if an early MSK1 upregulation could be revealed directly.
Agree. The experiments whose findings are reported here, were the last in a series of studies using MSK1-/- mice and WT littermates, and we have no more mice from that line. To start breeding the line again from in vitro fertilisation for this single experiment does not seem feasible either on ethical or financial grounds. cKO and WT littermate mice will be available in a few months, when we could check earlier time-points in a related line. However, those animals will have somewhat different genetic background from those we used in the present study. In addition, we are convinced that the data we present in the manuscript are interesting and important enough to get published without using an earlier time point.
In addition, the heading „MSK1 activation in SSDH by peripheral inflammation” should be reformulated.
Agree. Revised accordingly.
In the title, inflammatory pain is mentioned, but no pain responses were measured in the present study. The title should be reformulated.
Agree. The CFA model is widely used inflammatory pain model therefore we did not include any behaviour data. During the revision, we have added a figure showing that CFA injection leads to inflammatory pain, which is significantly attenuated in MSK1-/- mice.
In the case of Figs. 4 and 5, no statistics are mentioned, and no significance is indicated.
Agree. We are sorry for the lack of details on statistics and indication of significance in the figure. We have revised the manuscript accordingly.
I found no table in the manuscript, although the authors cite Tables 1-3.
Agree. We are sorry that this error has not been noticed during proofreading. Tables were part of the main material but then we moved them to Supplementary Materials. The manuscript has been revised accordingly.
Reviewer 2
The study by Irfan et al examines MSK1 expression and activation in the superficial spinal cord following CFA induced inflammatory pain. This histological study reveals that MSK1 is expressed in inhibitory and excitatory neurons, microglia and oligodendrocytes.
In general, the methods are adequate and the results are clearly reported. Overall, the study accomplished its main goal, by showing the specific cell types that express MSK1 in the spinal cord of wildtype mice and following CFA administration. It also assessed the spinal expression of activated MSK1 and one of its downstream effector S10H3 after CFA. Overall, a strong study with the following minor concerns noted.
1. The term ‘via firing up’ in the title conveys a more robust effect than that reported in the manuscript. The results showed CFA increased p-S10H3 at 1 timepoint but had no effect on pMSK. These effects could also be post-translational modifications. Furthermore, it is not demonstrated that S10H3 is only activated by MSK1.
Agree. The title implies more robust effect then we report. Accordingly, we have rephrased the title.
Regarding the role of MSK1 in S10H3 phosphorylation, we consider them as post-translational modification in the signalling cascade (e.g. MAPK – MSK1 – S10H3). The manuscript provide does provide evidence for the role of MSK1 in Supplementary Figure 3, which shows complete lack of p-S10H3 expression 1 hour after CFA expression in MSK1-/- mice’s spinal dorsal horn indicating the critical role of MSK1 in phosphorylating S10H3. This finding is in agreement with previously published data.
2. Three sections were used for quantification. It is unclear whether 3 sections/mouse were averaged or totaled. Also, the exact number of mice used in each group (and in the study) should be stated
Agree. We are sorry for the lack of clarity.
Quantified data are based on counting immuno-positive cells in 3-5 sections from each of three animals. The number of immuno-positive cells in each animal was averaged, and the average numbers were used for statistics and visualisation. The manuscript has been revised accordingly.
3. There is a discrepancy in the size of the sections, as reported Both 20 and 25 µm are stated.
Agree. The scale bars are different on two high magnification images.
The scale bars were set by ImageJ/Fiji. While the same objective was used taking the two raw images, the areas cropped from the original images by ImageJ/Fiji are different to fit into the respective figures, and scale bars were added after cropping. Hence, the scale bars are different. As the images are in different figures, we think it is not confusing.
4. It is concerning that for both p-MSK1 and p-S10H3, the number of cells expressing them is substantially lower than that expressing MSK1 (much lower for pMSK1). It could be assumed that the timing of the study (1 hour and 2 days) missed their critical period of activation, which can be before the1 hour timepoint. In fact, studies suggest that phosphorylation might peak at an earlier timepoint. Also, taking into consideration the formalin-related studies that were referenced, the phase 2 response typically peaks around 20-30 min. It seems that 1-hour timepoint will not accurately reveal the secondary pathways activated. This seems to be more plausible than any difference in nociceptive input following formalin versus CFA administration, as stated in the discussion. If the latter is the case, then it is more likely that CFA will cause a greater effect than formalin. The discussion provided on this result is a bit superficial, and could be expanded on.
Agree. Although a significant number of cells express MSK1 in the spial dorsal horn only a fraction shows sequel of MSK1 activation such as MSK1-dependent upregulation in p-S10H3 expression. The number of p-S10H3 immuno-positive nuclei 1 hour after CFA injection is comparable with those found after burn injury, carrageenan-, formalin- or capsaicin injection or electrical stimulation of peripheral branches of nociceptors up to 3 hours after of injury/inflammation.
Initially we hoped to find correlation between pain-related behaviour and S10H3 and MSK1 phosphorylation, which would have involved assessment of pain-related behaviour. That assessment could reliably be done one hour after the injection due to performing the injection under general anaesthesia. Therefore, no earlier time points have been used. The experiments whose findings are reported here, were the last in a series of studies, and by the end of the study we ran out of both MSK1-/- and WT mice. To start breeding the line again from in vitro fertilisation for this single experiment does not seem feasible either on ethical or financial grounds. However, as suggested, we have extended the Discussion on this issue during the revision.
Reviewer 2 Report
Comments and Suggestions for Authors
The study by Irfan et al examines MSK1 expression and activation in the superficial spinal cord following CFA induced inflammatory pain. This histological study reveals that MSK1 is expressed in inhibitory and excitatory neurons, microglia and oligodendrocytes.
In general, the methods are adequate and the results are clearly reported. Overall, the study accomplished its main goal, by showing the specific cell types that express MSK1 in the spinal cord of wildtype mice and following CFA administration. It also assessed the spinal expression of activated MSK1 and one of its downstream effector S10H3 after CFA. Overall, a strong study with the following minor concerns noted.
1. The term ‘via firing up’ in the title conveys a more robust effect than that reported in the manuscript. The results showed CFA increased p-S10H3 at 1 timepoint but had no effect on pMSK. These effects could also be post-translational modifications. Furthermore, it is not demonstrated that S10H3 is only activated by MSK1.
2. Three sections were used for quantification. It is unclear whether 3 sections/mouse were averaged or totaled. Also, the exact number of mice used in each group (and in the study) should be stated
3. There is a discrepancy in the size of the sections, as reported Both 20 and 25 µm are stated.
4. It is concerning that for both p-MSK1 and p-S10H3, the number of cells expressing them is substantially lower than that expressing MSK1 (much lower for pMSK1). It could be assumed that the timing of the study (1 hour and 2 days) missed their critical period of activation, which can be before the1 hour timepoint. In fact, studies suggest that phosphorylation might peak at an earlier timepoint. Also, taking into consideration the formalin-related studies that were referenced, the phase 2 response typically peaks around 20-30 min. It seems that 1-hour timepoint will not accurately reveal the secondary pathways activated. This seems to be more plausible than any difference in nociceptive input following formalin versus CFA administration, as stated in the discussion. If the latter is the case, then it is more likely that CFA will cause a greater effect than formalin. The discussion provided on this result is a bit superficial, and could be expanded on.
Author Response

(The authors gave the same response as above.)
